# Exploring meteorological droughts' spatial patterns across Europe through complex network theory

Domenico Giaquinto[2], Warner Marzocchi[1,2], and Jürgen Kurths[2,3,4]

[1]University of Naples 'Federico II', Naples, Italy.
[2]Scuola Superiore Meridionale, Naples, Italy.
[3]Potsdam Institute for Climate Impact Research, Potsdam, Germany.
[4]Department of Physics, Humboldt University, Berlin, Germany.

**Correspondence:** Domenico Giaquinto (domenico.giaquinto@unina.it)

**Abstract.** In this paper we investigate the spatial patterns and features of meteorological droughts in Europe using concepts and methods derived from complex network theory. Using Event Synchronization analysis, we uncover robust meteorological drought continental networks based on the co-occurrence of these events at different locations within a season from 1981 to 2020 and compare the results for four accumulation periods of rainfall. Each continental network is then further examined to unveil regional clusters which are characterized in terms of droughts' geographical propagation and source-sink systems. While introducing new methodologies in general climate networks reconstruction from raw data, our approach brings out key aspects concerning drought spatial dynamics, which could potentially support droughts' forecast.

## 1 Introduction

Droughts are among the most severe climate extremes, negatively affecting environments as well as societies and economies (Taufik et al., 2017; Feng et al., 2021; Doughty et al., 2015; Walthall et al., 2013). As reported by the World Meteorological Organization (WMO, 2021), they were one of the most impactful natural hazard in terms of human losses during the period 1979 – 2019, accounting for 34% of disaster related deaths. To make matters worse, droughts are expected to grow and become more severe in the near future due to human related climate change (Spinoni et al., 2021). Additionally, although different drought types respond differently to increasing greenhouse gas concentrations and distinctions among geographic regions should be made, there is high confidence that water stress is increasing globally according to the Sixth Assessment Report of the Intergovernmental Panel on Climate Change (IPCC) (Wehner et al., 2021).

Despite the high scientific interest and effort towards a better understanding of this crucial topic, there are still substantial discrepancies concerning the assessment of droughts' trends for specific regions. As for Europe, significant impacts have been extensively reported over the years, e.g. on ecosystems (Bastos et al., 2020), economy (Naumann et al., 2021), agriculture (Beillouin et al., 2020). This climate phenomenon is closely monitored and studied both under present and future climate conditions (Spinoni et al., 2016; Marinho Ferreira Barbosa et al., 2021). When it comes to future scenarios, some studies claim that the entire continent will suffer under the increase of droughts' frequency and severity (Spinoni et al., 2018; Cook et al.,

2020). Others are more cautious in drawing such a conclusion, pointing out to the critical role of internal climate variability and associated uncertainties (Zhao and Dai, 2017; Vicente-Serrano et al., 2021). There are multiple reasons for these incongruences.

First of all, droughts are characterized by ample spatial variability (Vicente-Serrano et al., 2021): sub-continental or sometimes even sub-national areas behave discordantly than their neighbors and different studies place emphasis on different territories (Spinoni J and P, 2016; Vicente-Serrano et al., 2021; Spinoni et al., 2018; Böhnisch et al., 2021). In addition, we lack a unique and standardized definition of regions: each scientific team outlines the sub-continental area of interest discretionally, following political or economic boundaries, according to the obtained results, or using external classifications (Spinoni J and

P, 2016; Spinoni et al., 2018, 2015). Although being completely understandable procedures, this makes comparisons among studies difficult, if not unfeasible.

On top of that, there is not a unique definition of drought. Depending on causes and impacts, droughts are classified into different types, i.e. meteorological, agricultural, hydrological and economic (Wilhite and Glantz, 1985; Seneviratne et al., 2012). These typologies are characterized by an increasing level of complexity, with impacts ranging from atmosphere to

land, ecosystems and even social and economic systems: an absolute definition could result in a misleading oversimplification (Lloyd-Hughes, 2014). Numerous indices have been proposed to classify, monitor and assess this complex climate event (Wei et al., 2021). Some are defined in terms of anomalies of a single variable, while, for the most complicated cases, multiple atmospheric variables are taken into account (Marinho Ferreira Barbosa et al., 2021). Each of them has its strengths and limitation (Mishra and Singh, 2010; Zargar et al., 2011; Wei et al., 2021; Mukherjee et al., 2018) behaving differently when

it comes to projections of future climate scenarios (Böhnisch et al., 2021; Vicente-Serrano et al., 2021). Using a hydrological and water-use model, Naumann et al. (2021) conclude that hydrological droughts' damages in Europe could strongly increase with global warming and cause a regional imbalance in future impacts. On the other hand, Vicente-Serrano et al. (2021) show that trends of meteorological droughts over Western Europe are statistically non-significant from a long term perspective.

Finally, the reference period used to identify the anomalies of the drought variable of interest, being it precipitation, soil

humidity, streamflow etc., represents another important source of inhomogeneity between different studies (Spinoni et al., 2017). Just as an example, Vicente-Serrano et al. (2021) use the period 1871–2018 to compute the Standardized Precipitation Index (McKee et al., 1993), while Spinoni et al. (2018) choose 1981–2010 to compute the frequency of drought events.

In this study we focus on meteorological droughts, defined as a period with a precipitation deficit in relation to the long-term average condition for a given region. Being simply related to precipitations, meteorological droughts are the basis for every

other drought class. A lack of precipitation may be the first precursor of soil humidity's reduction (agricultural drought) as well as the cause of stream flow shortage (hydrological drought) (Bevacqua et al., 2021).

The approach we propose in this study is based on complex networks and it is designed to address some of the criticalities described before, while detecting unknown spatial features of European meteorological droughts. Complex networks have been applied for many different purposes, e.g. to study social dynamics, power grids, epidemics and many others. They are

a powerful tool in systems composed of many units, where the interactions' structure is closely linked to system's dynamics (Boccaletti et al., 2006). Over the last two decades, complex network theory has been successfully applied to climate science too, proving to be an efficient method both to deepen our general understanding of various complex climate processes and to

predict the occurrence of extremes (Boers et al., 2019; Tsonis et al., 2006; Ludescher et al., 2021). Complex networks have been applied to droughts too. Ciemer et al. (2020) developed a forecasting scheme to predict the occurrence of meteorological droughts in the Central Amazon basin. In their recent paper, Konapala et al. (2022) develop a method to assess droughts' propagation in North America.

Our main objective here is to uncover spatial features of meteorological droughts in Europe, highlighting underlying mechanisms and patterns which could potentially support drought's forecast in the future. We aim at distinguishing regions in Europe whose main feature lies in drought occurrence and propagation's coherence. Identifying such territories could be of a great importance to further investigate this phenomenon within those areas where its characteristics are homogeneous. Indeed, droughts display a high spatial and temporal variability, and it is thus fundamental to study their evolution accounting for these irregularities to possibly lower uncertainties. Furthermore, with our model we are able to describe the average historical patterns in droughts' evolution which could be a starting point for future climate studies to identify the spatial tracks that are followed by this climate hazard, building a forecasting scheme.

Our study is based on climate complex networks and on the concept of Event Synchronization, a nonlinear statistical similarity method useful to determine the correlations among spatial locations in terms of event co-occurrences. Using these tools we are able to identify drought regions in Europe based on the process itself and not depending on any external classifications, bringing out key aspects concerning drought dynamics at a regional scale for different rainfall accumulation periods from 1981 to 2020, while introducing new methodologies in general climate networks reconstruction from raw data. The understanding and ability of describing droughts as a complex phenomenon is still in a preliminary stage, but climate complex networks prove to be a powerful tool to reveal hidden features of this climatic process.

The reminder of the paper is organized as follows: in Section 2 we describe the data and the methodology we adopted to construct the meteorological drought networks and in Section 3 we present the results. Further insights and details on the methods are shown in the Appendix.

## 2 Data and Methodology

We follow the procedure described in Ludescher et al. (2021) and Fan et al. (2021) to reconstruct our meteorological droughts' networks from data, using the Standardized Precipitation Index (SPI) (McKee et al., 1993; Edwards, 1997; Guttman, 1999). The SPI measures precipitation's anomalies at a given location, based on a comparison of observed total rainfall for a certain accumulation time interval of interest (denoted SPI-1, SPI-3, SPI-6, SPI-9 and SPI-12 months) with the long-term historic record for that period and specific area (EDO, 2020). SPI values are positive (negative) for greater (less) than the median precipitation. The SPI is provided by the European Drought Observatory (EDO, 2020), computed from the monthly precipitation data of the Global Precipitation Climatology Centre. We focus on the time interval 1981 – 2020 and analyze four different accumulation periods, namely the SPI-3, SPI-6, SPI-9 and SPI-12, building one continental network for each of these cases, assessing their similarities and differences. The SPI-1 month is not considered, since it may not be accurate for regions with

90   high probability of zero accumulated rainfall during one month (EDO, 2020), which is the case of some areas of northern Africa included in our study.

    Our spatial domain is defined based on the IPCC's sixth assessment report (Jian and Hao-Ming, 2021; Iturbide et al., 2020): the regions NEU, WCE and MED are taken as representation of Europe (Figure 1). Notice that meteorological droughts are defined on land and thus sea is not considered.

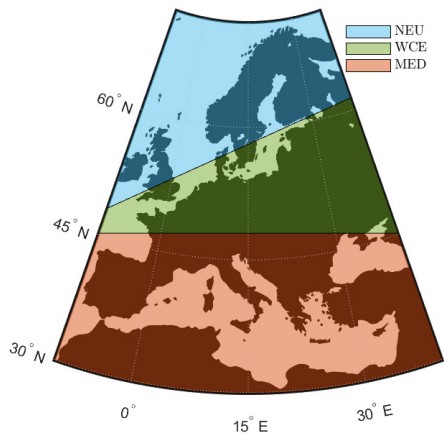

**Figure 1.** The spatial domain analyzed in this study using the IPCC regions NEU, WCE and MED (only land is considered). Latitude from $30°$ to $72.6°$ and Longitude from $-10°$ to $40°$.

95   The spatial domain is discretized with a resolution of $0.25° \times 0.25°$ (latitude x longitude). Each grid point is associated to four SPI series (one for each accumulation period) and will serve as node of the graph.

    To identify meteorological droughts in the SPI time series we refer to McKee's work (McKee et al., 1993), where the author proposes a functional and quantitative definition of drought using the SPI as indicator: a meteorological drought is an event that occurs whenever the value of the SPI $\leq -1$, whatever the accumulation period is. This definition and the related drought's

100   classes (Table 1) are still accepted and wildly used (Spinoni J and P, 2016; WMO, 2021). The event sequences from the complete SPI time series are recovered by selecting those occurrences with SPI $\leq -1$ and consecutive events are removed, keeping only the first of successive drought conditions (see Figure 2).

**Table 1.** Meteorological drought categories defined for values of the SPI from (McKee et al., 1993).

| SPI values | Drought category |
|------------|------------------|
| -1 to -1.49 | moderate drought |
| -1.50 to -1.99 | severe drought |
| $\leq$ -2 | extreme drought |

According to the event-like nature of this process, we use Event Synchronization (Quiroga et al., 2002; Malik et al., 2012) as statistical similarity measure to construct each of the four continental drought networks. Event synchronization is a powerful nonlinear method to assess the similarity of event series with not equal spacing between successive occurrences and thus it is especially appropriate for studying extreme events (Fan et al., 2021). The degree of synchronicity of two event series is measured based on the relative timings of events and it is obtained from the number of quasi-simultaneous occurrences. We summarize the advantages of this method as follows: (i) ES is designed to treat event-like time series; (ii) by using ES there is no need to set a specific time lag; (iii) ES has both a symmetric and asymmetric formulation, eventually being able to show driver-response relationships; (iv) ES has been extensively used as a tool to construct climate extreme events' networks, proving to be enough efficient and informative (Malik et al., 2010; Agarwal et al., 2017; Boers et al., 2019; Strnad et al., 2023). The detailed algorithm is described in Fan et al. (2021) and is shortly repeated in the Appendix for the convenience of the reader. The edges derived through Event Synchronization represent the synchronicity in drought's occurrences between the nodes of the graph. In this preliminary stage, we are not concerned with the direction of synchronicity: two location are linked if they display a co-evolution of meteorological drought events; therefore, the resulting four continental networks are undirected.

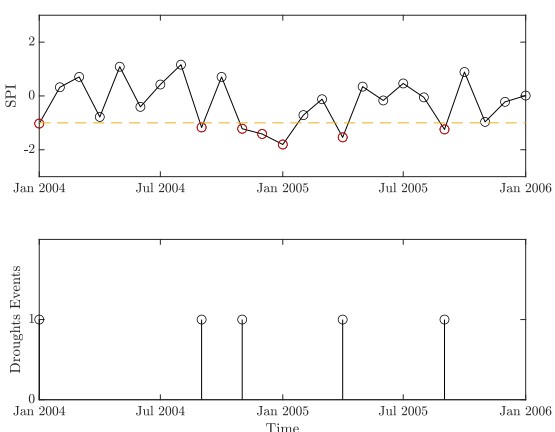

**Figure 2.** Construction of the Event Series. From the complete SPI Time Series (top panel) we select those realizations with SPI $\leq -1$ to form the Droughts Event Series (bottom panel). Notice that consecutive events are removed, keeping only the first occurrence. This procedure is carried for every node of the space domain and for the entire time period 1981 - 2020.

The connections contained in the four continental networks have different strengths, i.e. value of synchronicity degree (see equation A4). Given the large extension and the high resolution of these graphs, it is not essential to carry the information of links' weights: instead, it is more convenient to select only the strongest edges and derive unweighted representations of each network (see equation A5), which will still preserve the main structures of the original graphs.

Extracting the connections with the highest statistical similarity from the complete weighted graph is a very common procedure in general functional network reconstruction from raw data (Zanin et al., 2012), and it is well known in climate networks science too (Ludescher et al., 2021; Gupta et al., 2021; Ciemer et al., 2020). In this particular field, the strongest links are

considered to be the backbone structure of the climatic process under examination: indeed, when a climate correlation or synchronization network is built, it often resembles an all-to-all connected graph, giving no useful information about the main underlying structure which remains hidden under the great number of links. Cutting-off the weakest edges is thus a fundamental step to get a clearer and meaningful representation of any climatic process (equation A5). Still, we lack so far a systematic and well accepted procedure to select this cut-off threshold $\theta$ in A5 for climate networks, and, moreover, to assess the robustness of the selected connections. As a result, scientists tend to choose it discretionally on a case by case basis (Kurths et al., 2019), generally optimizing the ratio between high correlation and sufficient number of events for comparison (Stolbova et al., 2014). Here we propose a rigorous way to choose the lower bound of synchronicity below which a connection is discarded based on the Hamming distance (equation B1 and B2).

Instead of constructing one 1981-2020 drought network for each accumulation period, we divide the complete data-set into two sub-databases (one with data from the period 1981-2000 and one with data from 2001-2020) and build two independent sub-networks; we then choose the cut-off threshold which minimizes the Hamming distance between these two sub-networks (Figure B1). With this procedure we not only identify the best cut-off threshold, but we have the additional advantage of selecting the connections which are significant and robust in time because they arise from independent sets of data: the network so reconstructed does indeed retain the long lasting backbone structure of the process, discarding the majority of the spurious synchronizations caused by internal variability and noise. More details about this method are given in the Appendix.

After having built the four unweighted and undirected European networks, we proceed in partitioning them into regional clusters. This procedure is followed in Konapala et al. (2022) as well, where the authors use a distance weighted synchronization to construct the graph. This filter strengthens the synchronicity among close locations, while penalizing any potential connection between distant areas. Moreover, the distance weight introduced by Konapala et al. (2022) is a relative metric, tuned on the extent of the studied domain. Even assuming that drought conditions' propagation is influenced by geographical distance and therefore bounded to a certain level, this sensitivity should at least be fixed, not changing with the size of the studied area. Whereas according to the majority of studies in climate networks long connections can not be ignored; on the contrary, teleconnections play a crucial role in transferring climate information across the globe (Donges et al., 2009; Tsonis et al., 2008, 2006; Boers et al., 2019, 2013; Stolbova et al., 2014). These considerations prompts us to identify regional clusters from the Event Synchronization network without introducing any distance weight.

To find the regional clusters for each of the four networks, we use the Louvian algorithm (Blondel et al., 2008), a heuristic method that is based on modularity optimization (Newman, 2006). The Louvian algorithm finds a local maximum of modularity, which depends on the order the nodes are picked. Therefore, we apply it 10,000 times for each continental network, taking the partition with the overall highest modularity. The found communities represent the regions in which a drought event is more likely to propagate once started: intra-connections are maximized over inter-connections, meaning that the nodes grouped together are characterized by a high cooperativity in terms of synchronization in meteorological droughts' occurrences.

To analyze the synchronization's patterns of meteorological droughts, we build regional spatial networks for each community. This time, unlike what was done before for the four continental networks, once the nodes of one specific community are extrapolated, we use Event Synchronization to build weighted and directed graphs.

The comparison between the four continental networks and the features of the regional networks are described in the next Section.

## 3 Results

The partitions of the four European drought networks resulting from 10,000 different realizations of the Louvian algorithm are shown in Figure 3.

We notice important similarities and differences between the four networks. The most relevant features that arise are the following: i) the regional clusters of the Scandinavian Peninsula are quite comparable, with the western part of Norway always standing alone as one community, separated from Sweden, which forms another compact block, and Finland, often divided into several parts alongside the Kola Peninsula; ii) the eastern part of the continent is split into latitudinal regions, well visible for the shortest accumulation periods while more and more fragmented in the 9 and 12 months cases; iii) Turkey often forms one cluster; iv) the Iberian peninsula is connected to North-west Africa, except for the SPI-12 network; v) the norther part of Italy is joined with the north of the Balkan Peninsula; vi) the increase of the fragmentation of the communities with the accumulation period does not translate into an increment of the communities' number (see Table 2), but to the disruption of the spatial continuity of the clusters from the shortest accumulation periods to the longest one.

We find that the latter characteristic is due to the presence of long links, more numerous the higher the accumulation period: we register 25565 long links in the SPI-12 Network, one order of magnitude more than the 2722 long links in the SPI-3 Network (see Figure 4 and Table 2). In Boers et al. (2019) the extreme precipitations' global climate network was characterized by two different weather systems, a regional power-law distributed one and a super-power-law-distributed global pattern. This latter system was detected from network's links longer than 2500 kilometers. In our analysis, these features seem to emerge again, even if we focus on negative precipitation's extremes and Europe is not big enough to contain a statistically adequate number of possible long connections ($\geq$ 2500 km). Nevertheless, if we look at Figure 4, we see a shift in the distribution of the number of links when this critical 2500 km length is passed. Moreover, the number of long links increases with the accumulation period, even when the total number of connections does not change sensibly among the four cases (see Table 2).

**Table 2.** Number of total and long links ($\geq$ 2500 km) in the four Europe's meteorological drought networks and number of found communities.

|  | Total Links | Long Links ($\geq$ 2500 km) | % of long links | Communities |
|---|---|---|---|---|
| SPI-3 Network | 2340806 | 2722 | 0.12 | 19 |
| SPI-6 Network | 2210586 | 7074 | 0.32 | 19 |
| SPI-9 Network | 1979005 | 14599 | 0.74 | 19 |
| SPI-12 Network | 1615415 | 25565 | 1.58 | 21 |

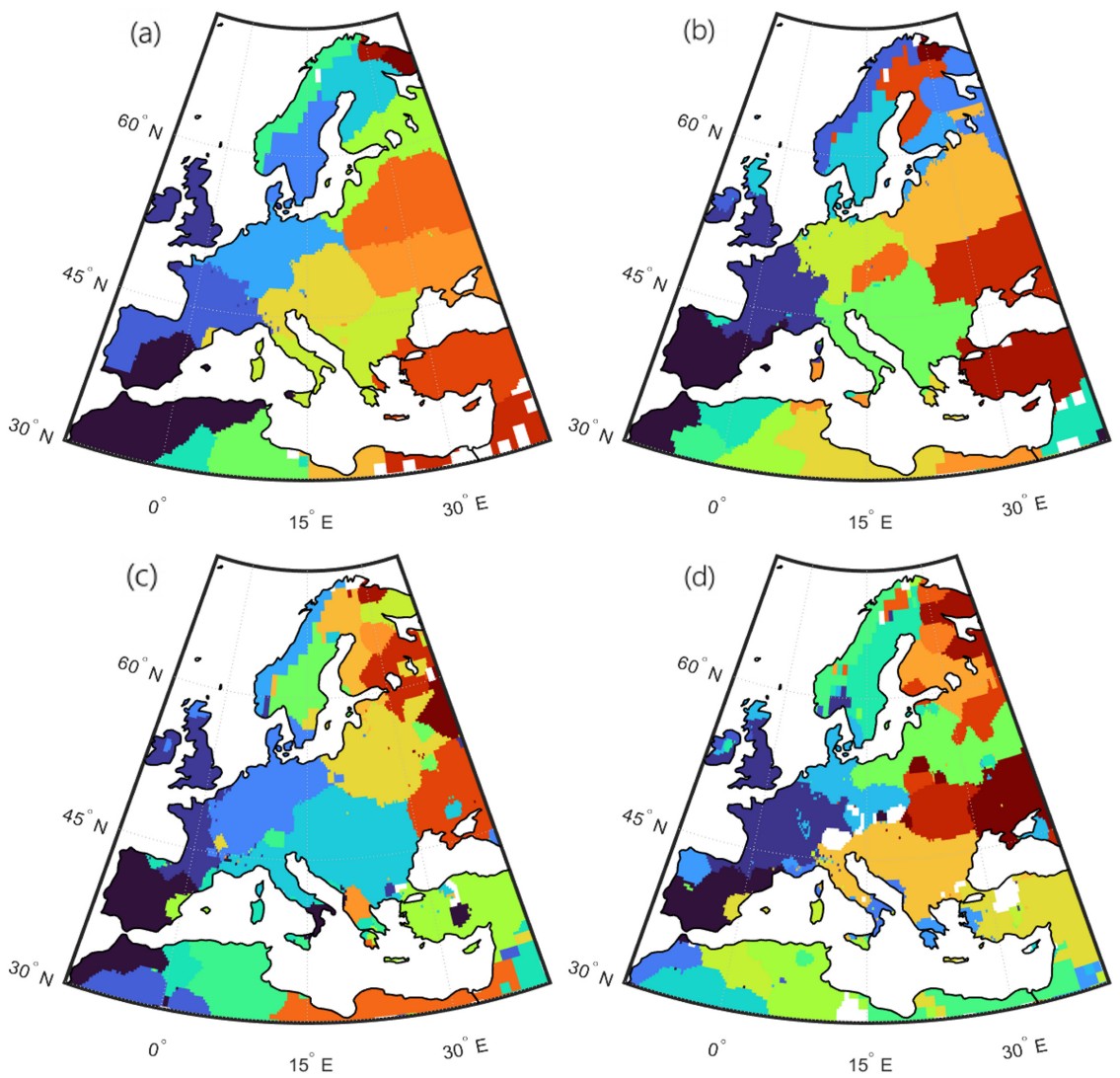

**Figure 3.** Communities detected with the Louvian algorithm from the SPI-3 Network (a), SPI-6 Network (b), SPI-9 Network (c) and SPI-12 Network (d). Each color represents a different community while white areas are either part of the sea or consist of nodes belonging to clusters with less than 100 nodes.

This exhibits the primary role of long connections in the context of climate networks. The presence of these long links is typically due to large scale atmospheric patterns which could act as common drivers of climate extremes of distant regions. This feature would not emerge if we used a distance weighted network's construction: in fact, we would have lost long connections, being left with four partitions very similar one to the other. We argue that short meteorological drought events (with accumulation period up to 6 months) are driven by regional climate systems, while long ones (with accumulation period from 9 months onward) are due to large scale patterns that affects ample portion of the continent. Rossby wave trains could be a

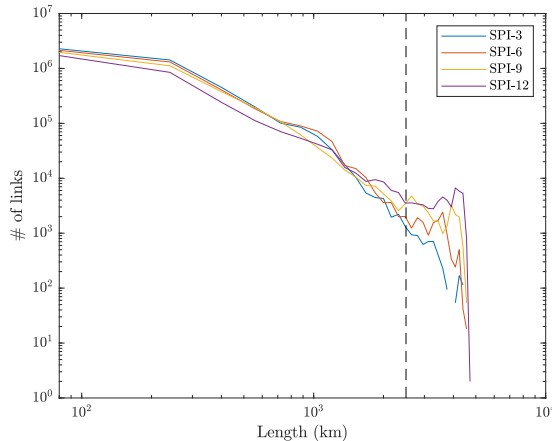

**Figure 4.** Distribution of the number of links with their length (great circle distance of the two nodes connected by the link) for each of the four Europe's meteorological drought networks (colored lines). The dashed line represents the 2500 km distance.

potential reasonable candidate in connecting distant area in longitudinal direction. Following studies could better clarify the climate precursors responsible for the appearance of these long connections for high accumulation periods.

Once we identify the clusters of each of the four continental networks, we proceed in studying each regional community separately, constructing weighted and directed graphs. The antisymmetrical score $q_{ij}$ (Equation A4) which results from the application of Event Synchronization (see the Appendix) to the regional clusters can be used to derive the out-degree $k_i^{out}$ (defined as the average of the weights of the outgoing links) and the in-degree $k_i^{in}$ (defined as the average of the weights of the incoming links) of each node $i$ in its community $c$ and, consequently, its overall degree centrality $k_i$:

$$k_i = k_i^{out} + k_i^{in} = \frac{1}{N_c - 1} \sum_{j \in \mathcal{N}_i} q_{ij} + \frac{1}{N_c - 1} \sum_{j \in \mathcal{N}_i} q_{ji} \tag{1}$$

where $N_c$ is the total number of nodes belonging to community $c$ and $\mathcal{N}_i$ are the neighbors of node $i$. The degree centrality $k_i$ lies between $-1$ and $1$. Here, a node $i$ with $k_i < 0$ is defined as a sink, otherwise as a source. Each community of each of the four continental networks can be thus studied under this framework, identifying a source-sink system.

We show here some interesting source-sink systems found in each of the four continental networks.

In the SPI-3 network Portugal essentially acts as a source for its community (Figures 5a and 5b), suggesting that the lack of humidity could move from the coast towards inland. The role of Portugal as source could be explained by two main reasons: on the one hand, this area is on average the rainiest in the Iberian Peninsula and thus it is more sensitive to dry conditions; on the other side, it is more affected by the North Atlantic Oscillation and the East Atlantic pattern, two important atmospheric processes which influence the Iberian precipitation regime (Benito et al., 1996). A different situation is depicted in the central Europe community (Figures 5c and 5d), where the internal part of the region drives droughts' occurrences to the coast. As shown by Hofstätter et al. (2018), precipitation patterns over central Europe are largely controlled by atmospheric cyclones:

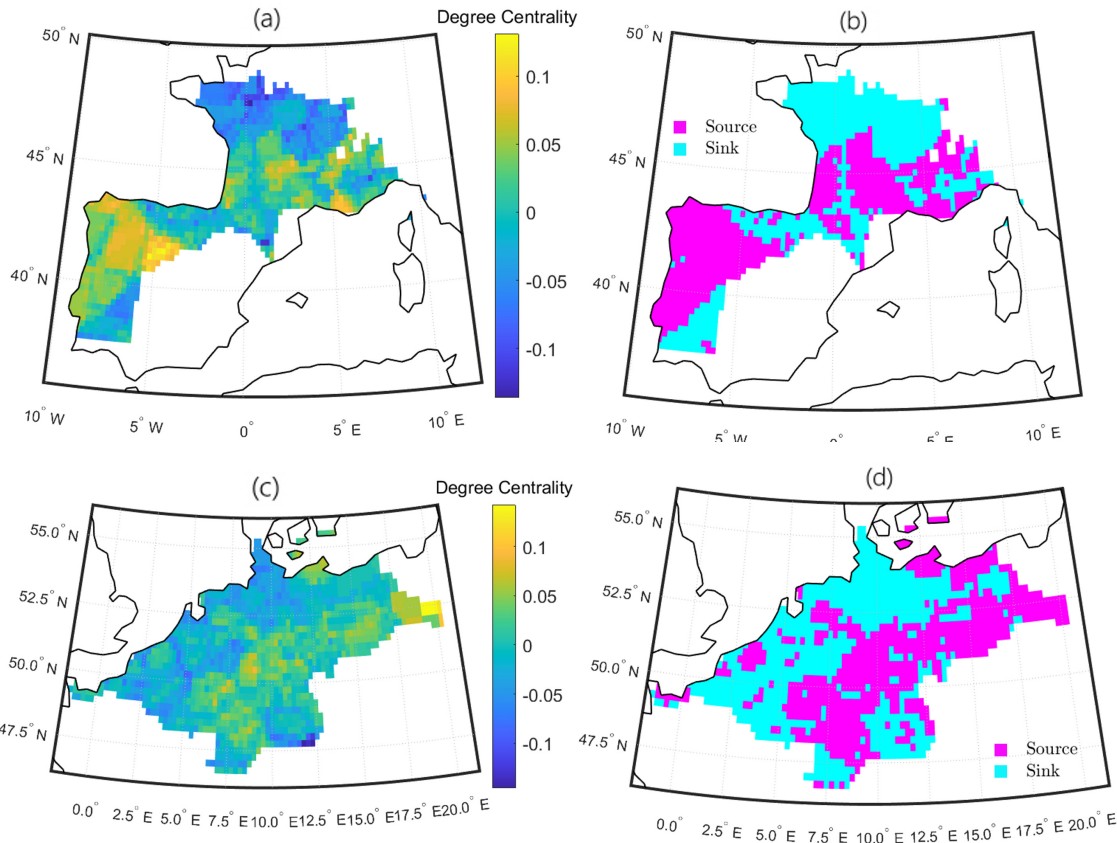

**Figure 5.** Two regional spatial networks from the SPI-3 Europe meteorological drought network. Degree centrality (a, c) and source-sink system (b, d).

consequently, the evolution of meteorological droughts in this region may be directed along cyclones' tracks. A further investigation into the average patterns of the various cyclone types may help in clarifying better this matter.

In the SPI-6 networks, it is interesting to notice the connection between northern Ireland and the Norwegian mountains (Figures 6a and 6b). As we already pointed out looking at Figure 3, the western part of Norway is always separated from the rest of the Scandinavian peninsula in every of the four accumulation periods networks, but in the specific case of the SPI-6 network, northern Ireland is synchronized with this mountainous chain and in particular it uniformly acts as a sink. While it seems clear that the separation of Norway from the rest of the Scandinavian Peninsula arises from the blocking action of the Norwegian mountains, the linkage between northern Ireland and Norway at this accumulation period is unforeseen; nevertheless we anticipate the prominent role of atmospheric rivers moving through the Norwegian sea and the Scandinavian pattern, which leads to dry conditions over the northern part of the continent during its positive phase (Bueh and Nakamura, 2007). Another regional spatial network derived from the SPI-6 graph that we show here is the Turkey one. We mentioned that

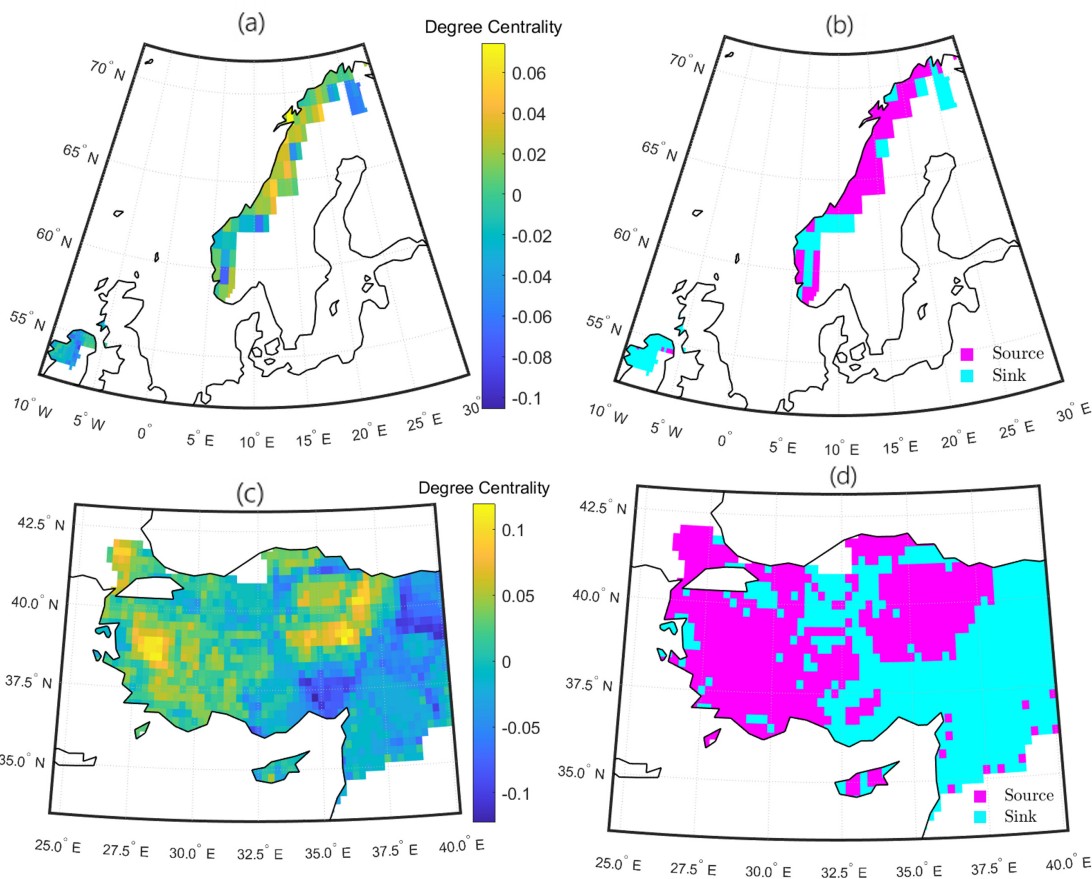

**Figure 6.** Two regional spatial networks from the SPI-6 Europe meteorological drought network. Degree centrality (a, c) and source-sink system (b, d).

this region often consistently form one sole cluster, displaying a certain stability over the different accumulation periods cases. In the SPI-6 network the western and central regions of Turkey precede the East in the occurrence of meteorological droughts events (Figures 6c and 6d). While the North Atlantic Oscillation represents a remote driver of precipitations over Turkey, the
220   Mediterranean and the marine Polar air masses are direct causes of rainfall in this region (Sariş et al., 2010), and their west-east direction suggests a key role in droughts' diffusion as well.

For the SPI-9 network we show two communities which are somehow complementary to each other (Figure 7): the first one (Figures 7a and 7b) includes the majority of the Iberian peninsula with a small region of Turkey while in the second one (Figures 7c and 7d) we see the remaining part of Turkey with a small region of Spain. The only difference between the two
225   communities, which may cause the distinction of this two regional clusters, is the fact that in the first case a portion of northern Africa and south Italy come into the picture as well. Looking at these two examples we highlight two main features: i) Portugal is still a big source (7a and 7b), as happened in the SPI-3 case, and ii) long connections strongly contribute in shaping the

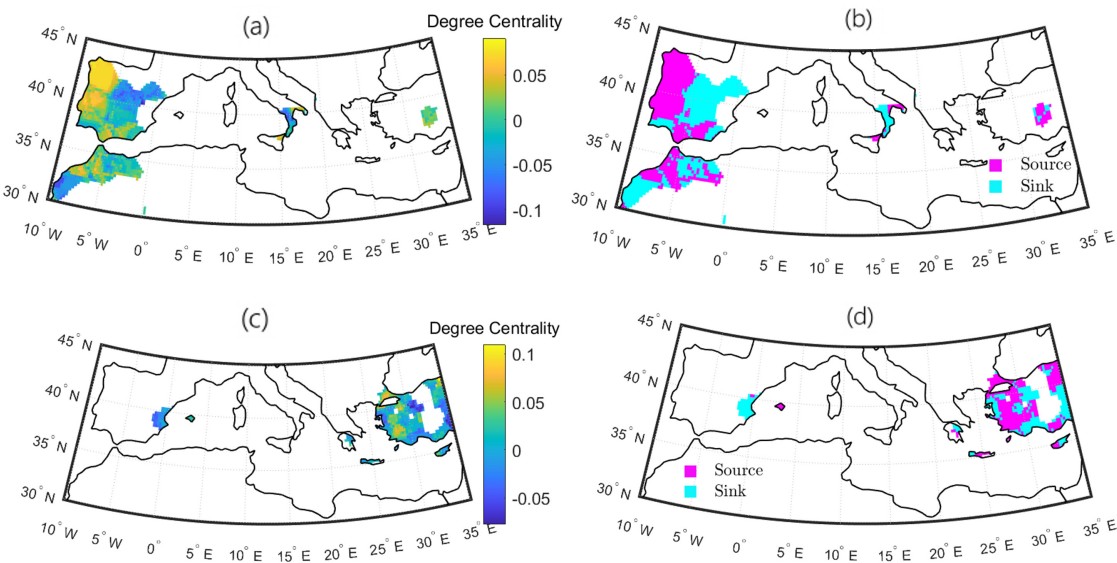

**Figure 7.** Two regional spatial networks from the SPI-9 Europe meteorological drought network. Degree centrality (a, c) and source-sink system (b, d).

clusters' landscape, linking distant regions which are not generally related in the short accumulation periods' networks. This latter characteristic is even accentuated in the SPI-12 communities structures. We show in Figure 8 two examples taken from this last network. In the first one (Figures 8a and 8b) we see a community very similar to the one seen in Figure 7c: the majority of Turkey is still connected to the same nodes of eastern Spain, but, differently from before, this time the Spanish region is a source for the rest of the community. This shows that the same nodes may play different roles in different accumulation periods: in fact, they may belong to different communities or, as this latter examples show, they can still be part of similar clusters but with a shift in the proportion of incoming links versus outgoing ones. As we already pointed out previously, the presence of long links in higher accumulation period networks contributes in connecting distant regions in longitudinal direction more than latitudinally, and the reason may lie in the propagation of Rossby waves. In Figures 8c and 8d there is a clear vertical separation in the source-sink system: the west part of both Great Britain and France is a source for the eastern regions of these two territories. The cluster also includes a small region of the Scandinavian peninsula. The role of western France as source for the eastern part of the country seems to be confirmed by the increasing trend in meteorological drought events in this region, as reported in Spinoni et al. (2016) too. Vidal et al. (2010) also show that the majority of meteorological drought events at the 12 months' timescale is located over the south and the eastern coast of France. As for Great Britain, the north and the central and southern part of the country consistently belong to different clusters in the SPI-6, 9 and 12 networks (see Figure 3). This could be related to the NAO's impact in the UK, with positive correlation to precipitation in the north and negative in the south (Rahiz and New, 2012). Moreover, the source's role of Wales and south England displayed here (Figure 8d) is consistent with previous studies (Phillips and McGregor, 1998; Fowler and Kilsby, 2002).

Finally, we point out that every region has its own precipitation regime, which is in turn affected by different atmospheric processes and patterns, whose influence also changes according to the specific time scale. For this reason, each of the showed regional source-sink system should be studied separately in the future in more detail.

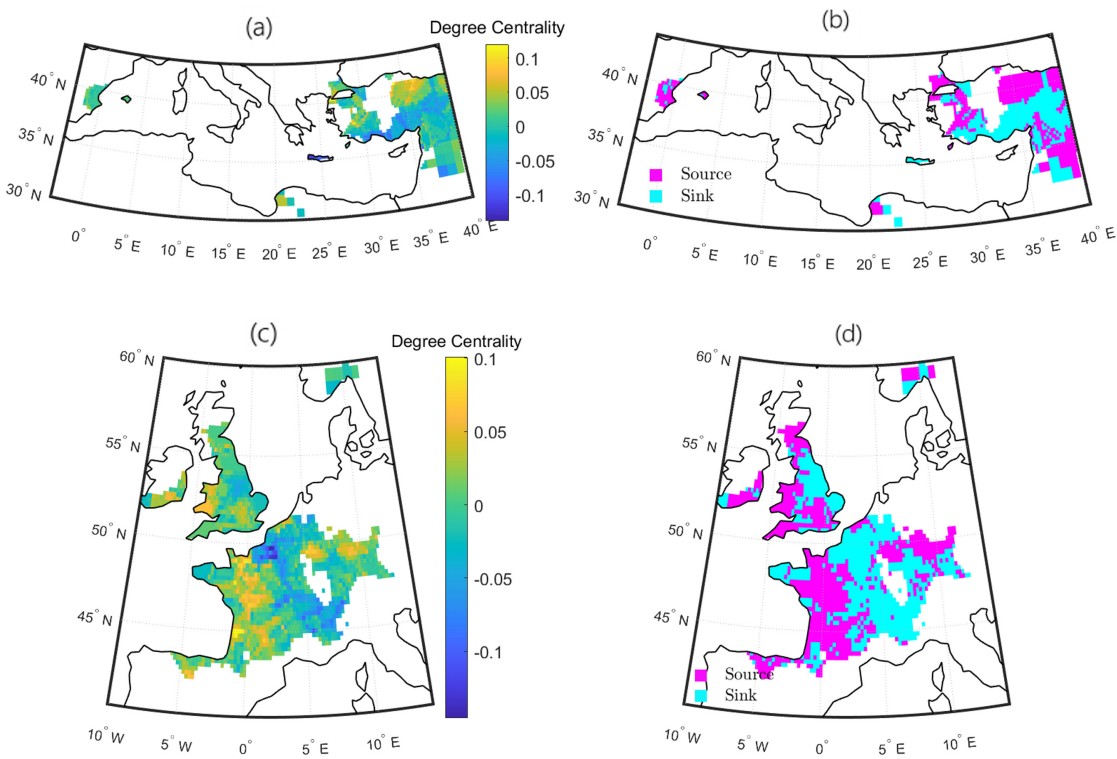

**Figure 8.** Two regional spatial networks from the SPI-12 Europe meteorological drought network. Degree centrality (a, c) and source-sink system (b, d).

## 4 Conclusion

In this study we propose a method to build robust climate networks from data and to identify meteorological drought regions and related source-sink systems in Europe.

Our networks are based on the synchronicity of drought's occurrences within a season and are constructed based on the Standardized Precipitation Index for four accumulation periods, 3, 6, 9 and 12 months, highlighting similarities and differences between those. The features of meteorological droughts for different timescales have never been investigated before through the lens of network theory. Here, we find that long connections play a crucial role in shaping drought regions, and become more and more important when long accumulations are considered. This suggest that, while short periods of precipitations' deficiency (1 to 6 months) tend to evolve in confined regions, long lasting droughts (9 to 12 months) are likely to propagate

across large portions of the continent. Short meteorological drought events could be driven by regional climate systems, while long ones by large scale patterns, like Rossby waves, but further studies are needed to better clarify the atmospheric precursors

of these long connections.

The Hamming procedure that we introduce here offers a new way to reduce the uncertainty of link attribution in unweighted and undirected networks reconstruction from raw data, when there is no previous knowledge about the system under study. This method could be potentially applied to different areas of study, and also be extended to weighted and directed graphs. From these more reliable continental networks, we uncover regional communities by applying the Louvian algorithm, a well

recognized method for cluster identification. We believe that this way of partitioning a certain geographical area via network theory concepts could be helpful in climate region identification, shaping them not via external classification or general climatic variables, but based on the specific climate process under study.

Meteorological drought regions are separately studied identifying source-sink systems. These systems highlight the spatial patterns along which the drought events have developed on average during the period 1980 - 2020. Moreover, they could be a

useful tool to use in forecasting these events in the sink nodes when the sources experience a drought condition.

## Appendix A: Event Synchronization networks.

Starting from two event series $i$ and $j$, an event $l$ that occurs at $i$ at time $t_l^i$ is considered to be synchronized with an event $m$ that occurs at $j$ at time $t_m^j$ if $0 < t_l^i - t_m^j \leq \tau_{lm}^{ij}$, where

$$\tau_{lm}^{ij} = min\{t_{l+1}^i - t_l^i, t_l^i - t_{l-1}^i, t_{m+1}^j - t_m^j, t_m^j - t_{m-1}^j, 8\}/2. \tag{A1}$$

Notice that we set a maximum time lag of 4 months above which the possible synchronizations between two nodes are disregarded. This means that the synchronicity between two locations is considered as such if it happens at most in the timescale of a season.

A score $J_{lm}^{ij}$ is assigned for every couple of events $l$ and $m$ in $i$ and $j$ according to the following rule:

$$J_{lm}^{ij} = \begin{cases} 1, & \text{if } 0 < t_l^i - t_m^j \leq \tau_{lm}^{ij}, \\ 1/2, & \text{if } t_l^i = t_m^j, \\ 0, & \text{else.} \end{cases} \tag{A2}$$

Finally, the number of times an event appears in $i$ shortly after it occurs in $j$ is counted

$$c(i|j) = \sum_{l=1}^{L} \sum_{m=1}^{M} J_{lm}^{ij}, \tag{A3}$$

where $L$ and $M$ are the total number of events in $i$ and $j$ respectively. The symmetrical $Q_{ij}$ and the antisymmetrical $q_{ij}$ synchronization scores are then defined as follows

$$Q_{ij} = \frac{c(i|j) + c(j|i)}{\sqrt{ML}}, \qquad q_{ij} = \frac{c(i|j) - c(j|i)}{\sqrt{ML}}. \tag{A4}$$

The symmetrical score is suited for undirected networks while the antisymmetrical one contains the additional information of the overall direction of the synchronicity between $i$ and $j$: if there is a majority of events happening at $i$ after they appear at $j$ compared to how many develop at $j$ after taking place at $i$, then $q_{ij}$ will be positive, with link's direction from $j$ to $i$. Viceversa, the sign will be negative, with direction of the link from $i$ to $j$. The value of the antisymmetrical score represents the percentage of times the $j$'s events precede $i$'s (or viceversa, depending on the sign), and serve as weight of the link. It can be interpreted as

the frequentist probability of observing an event happening at $i$ (during the next four months) given that an event has happened at node $j$. It is possible to organize the synchronicity measures between each pair of nodes in an $N \times N$ matrix, where $N$ is the number of nodes our spatial domain is discretized into. Each entry of the matrix lies between $0$ and $1$ and represent the strength of synchronization of node $i$ and $j$. In principle, the synchronization between a node with itself is equal to $1$, but we set it to $0$ since self-loops are not meaningful in this context. We use $q_{ij}$ for the weighted and directed community-specific networks,

while $Q_{ij}$ is used for the four continental graphs. In this latter case, we even pass to an unweighted representation, recovering the adjacency matrix $A_{ij}$ by preserving only the links with a synchronization score $Q_{ij}$ greater than a certain threshold $\theta$:

$$A_{ij} = \begin{cases} 1, & \text{if } Q_{ij} \geq \theta \\ 0, & \text{else.} \end{cases} \tag{A5}$$

    The choice of the threshold is conducted via the Hamming distance procedure.

**Appendix B: Hamming distance procedure.**

We divide the complete 1981-2020 data-set into two sub-databases, one from 1981 to 2000 and another from 2001 to 2020. Then, we build one sub-network for each of the two time interval and for every possible cut-off threshold in terms of percentage of strongest preserved links (from 1 to 100). Notice that the two sub-databases are independent from each other, deriving from observations sampled in different periods of time.

    Hence, we obtain two hundred different Europe's undirected and unweighted drought networks. Since our aim is to build

a single network as representative as possible of Europe's base meteorological drought conditions for the whole period 1981-2020, we assume the differences between the two sub-networks 1981-2000 and 2001-2020 to be minimum, since they are reproducing the same processes using independent data. In other words, if two locations $i$ and $j$ are drought-synchronized, this should results in both sub-networks. The underlying assumption we make here is that even if the frequency and duration of meteorological drought events may be changing due to climate change, the physical mechanism that shapes the locations'

synchronization in drought events occurrences does not change in time and the resulting spatial structure modeled with the climate network remains reasonably stable.

    For every cut-off threshold we count the differences between the two sub-networks via the Hamming distance, which measures the global probability of non-equal entries in the two adjacency matrices:

$$H(A^{1981-2000}, A^{2001-2020}) = \frac{1}{N^2} \sum_{i,j} XOR(A_{ij}^{1981-2000}, A_{ij}^{2001-2020}), \tag{B1}$$

where

$$XOR(A_{ij}^{1981-2000}, A_{ij}^{2001-2020}) = \begin{cases} 1 & \text{if } A_{ij}^{1981-2000} \neq A_{ij}^{2001-2020} \\ 0 & \text{else.} \end{cases} \tag{B2}$$

The procedure is depicted in Figure B1. The Hamming distance is not the only possible choice to compare the topology of two networks (Zager and Verghese, 2008; Ullmann, 1976; Fernández and Valiente, 2001). The reason why we have chosen this metric over the others is threefold: the networks we are comparing are undirected, unweighted and have the same link

density. For this specific case, this method gives an enough informative result, being also computationally very easy. For more complicated scenarios, one should look into other more refined Graph edit distances (Gao et al., 2010) or into spectral methods (Jurman et al., 2011) to capture the changes of a graph as a whole.

Even if we expect the two sub-networks to be similar for each cut-off threshold, they can never be identical due to noise and climate's internal variability. Therefore, we select as definitive cut-off threshold $\theta_{def}$ the one for which the Hamming distance

is below 5%:

$$\theta_{def} : H(\theta_1) < H(\theta_2) < ... < H(\theta_{def}) \leq 0.05 < H(\theta_{def+1}). \tag{B3}$$

This requirement is satisfied for each of the four continental networks if we preserve the top 4% of links (Figure B2). Consequently, an unweighted and undirected link is placed every time the value of synchronization between two nodes is above the 96-th percentile. We want this threshold to be the same for the four cases in order to compare the resulting networks,

which are this way characterized by approximately the same link density.

The definitive continental networks are therefore so constructed for each of the four accumulation periods: (a) two undirected and unweighted sub-networks are retrieved from the data 1981-2000 and 2001-2020 respectively, taking the 96-th percentile as cut-off threshold and (b) the two sub-networks are intersected, taking only the common links between the two to form the final 1981-2020 Europe meteorological drought network.

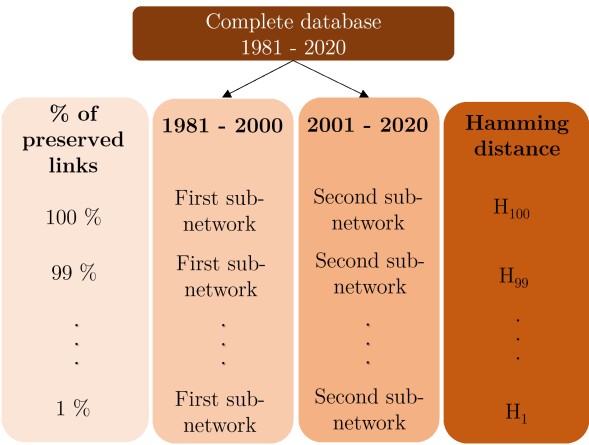

**Figure B1.** Procedure to select the percentage of strongest links to preserve. The original database is split symmetrically into two sub-databases. For each of them and for every cut-off threshold an undirected and unweighted Event Synchronization network is constructed. Every pair of sub-networks is used to compute the Hamming Distance.

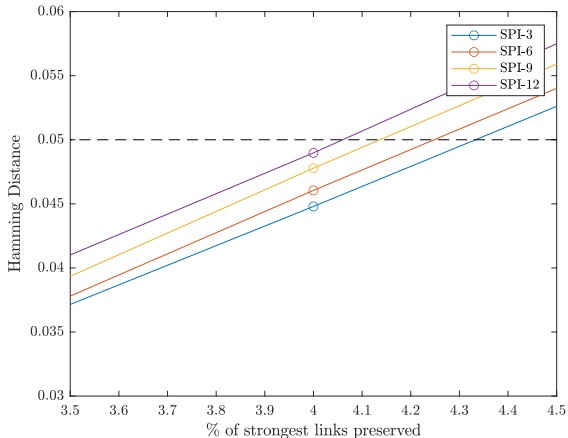

**Figure B2.** The Hamming Distance as a function of the percentage of strongest link preserved. By choosing the 96-th percentile as cut-off threshold (preserving the top 4% of links) the differences between the two sub-networks are below 5% in each of the four cases.

*Author contributions.* All authors contributed to planning the work and writing and editing the manuscript.

*Competing interests.* The authors declare that they have no conflict of interest.

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
