# Peer review of "Exploring meteorological droughts' spatial patterns across Europe through complex network theory"

_Nonlinear Processes in Geophysics, 2023_

## Referee Comment (RC2)

**Comments to the manuscript: Exploring meteorological droughts' spatial patterns across through complex network theory**

In the paper, Event Synchronization analysis are applied to investigate the spatial patterns and features of meteorological droughts in Europe. The main points of the manuscript is expressed concisely and the paper is overall well organized. The standard of English is acceptable. Here are a few comments and suggestions.

1. In Introduction, the research motivation of the spatial patterns and features of meteorological droughts is not clear enough.

2. Why is Hamming distance used to calculate the difference between two subnetworks, is there any other distance (measure) can be used.

3. In the study of the spatial pattern and characteristics of meteorological drought, what are the advantages of using method Synchronization analysis compared to other methods.

---

## Author Comment (AC1)

We thank the referee for investing her/his valuable time to help us improve our work with her/his accurate suggestions. We have revised our manuscript, taking into consideration all the referee's comments. The parts in red have been added to the manuscript to improve our work.

**1. Poor quality of the figures (maps): the maps are too small, with land boundaries difficult to visualise (particularly when covered by dark colours); for the maps in figures 5 to 8, the colour scale should have a label indicating what it is representing (ki).**

We increased the figures' size and improved the general appearance as suggested, including the labels on the colorbars in the Figures 5 to 8 of the manuscript. Please see the following Figure 1 to get an impression. The manuscript has been modified accordingly.

[Figure]

Figure 1: Appearance of map figures before (left) and after the revision (right).

**2. The physical interpretation of the results would significantly enhance the relevance and quality of the paper; an attempt to identify the regional climate systems associated to the identified patterns, and an interpretation of the patterns based on atmospheric transport processes would be beneficial. What would explain the distinct behaviour of the western Iberia coastal areas and Biscay/France coastal zones?**

Every region has its own precipitation regime, which is in turn affected by different atmospheric processes and patterns, whose influence also changes according to the specific time scale. For this reason, we point out that each of the showed regional source-sink system should be studied separately in detail from a climatic perspective, crossing expertises from different fields, which is beyond the scope of this study. Nevertheless, we revised our work with possible explanation of the patterns we found.

1. SPI-3 regional spatial networks.

   Figure 5a and 5b of the manuscript (Figure 2 here):

   The role of Portugal as source could be explained by two main reasons: on the one hand, this area is on average the rainiest one in the Iberian Peninsula and thus it is more sensitive to dry conditions; on the other side, it is strongly affected by the North Atlantic Oscillation and the East Atlantic pattern, two important atmospheric processes which influence the Iberian precipitation regime [1].

[Figure]

Figure 2: Two regional spatial networks from the SPI-3 Europe meteorological drought network. Degree centrality (a) and source-sink system (b).

   Figure 5c and 5d of the manuscript (Figure 3 here):

   As shown by [4], precipitation patterns over central Europe are largely controlled by atmospheric cyclones: consequently, the evolution of meteorological droughts in this region may be directed along cyclones' tracks. A further investigation into the average patterns of the various cyclone types may help in clarifying better this matter.

2. SPI-6 regional spatial networks.

[Figure]

Figure 3: Two regional spatial networks from the SPI-3 Europe meteorological drought network. Degree centrality (c) and source-sink system (d).

Figure 6a and 6b of the manuscript (Figure 4 here):

While it seems clear that the separation of Norway from the rest of the Scandinavian Peninsula arises from the blocking action of the Norwegian mountains, the linkage between northern Ireland and Norway at this accumulation period is unforeseen; nevertheless we anticipate the prominent role of atmospheric rivers moving through the Norwegian sea and the Scandinavian pattern, which leads to dry conditions over the northern part of the continent during its positive phase [2].

[Figure]

Figure 4: Two regional spatial networks from the SPI-6 Europe meteorological drought network. Degree centrality (a) and source-sink system (b).

Figure 6c and 6d of the manuscript (Figure 5 here):

While the North Atlantic Oscillation represents a remote driver of precipitations over Turkey, the Mediterranean and the marine Polar air masses are direct causes of rainfall in this region [7], and their west-east direction suggests a key role in droughts' diffusion as well.

[Figure]

Figure 5: Two regional spatial networks from the SPI-6 Europe meteorological drought network. Degree centrality (c) and source-sink system (d).

3. SPI-9 and SPI-12 regional spatial networks.

Figure 7a, 7b, 7c, 7d and 8a and 8b of the manuscript (Figure 6 and 7 here):

As we already pointed out previously, the presence of long links in higher accumulation period networks contributes in connecting distant regions in longitudinal direction more than latitudinally, and the reason may lie in the propagation of Rossby waves.

[Figure]

Figure 6: Two regional spatial networks from the SPI-9 Europe meteorological drought network. Degree centrality (a,c) and source-sink system (b,d).

Figure 8c and 8d of the manuscript (Figure 8 here):

The role of western France as source for the eastern part of the country seems to be confirmed by the increasing trend in meteorological drought events in this region, as reported in [8] too. [9] also show that the majority of meteorological

[Figure]

Figure 7: Two regional spatial networks from the SPI-12 Europe meteorological drought network. Degree centrality (a) and source-sink system (b).

drought events at the 12 months' timescale is located over the south and the eastern coast of France. As for Great Britain, the north and the central and southern part of the country consistently belong to different clusters in the SPI-6, 9 and 12 networks (see Figure 3 of the manuscript). This could be related to the NAO's impact in the UK, with positive correlation to precipitation in the north and negative in the south [6]. Moreover, the source's role of Wales and south England displayed here is consistent with previous studies [3, 5].

[Figure]

Figure 8: Two regional spatial networks from the SPI-12 Europe meteorological drought network. Degree centrality (c) and source-sink system (d).

Finally, we point out that every region has its own precipitation regime, which is in turn affected by different atmospheric processes and patterns, whose influence also changes according to the specific time scale. For this reason, each of the showed regional source-sink system should be studied separately in the future in more detail.

**3. Minor comments.**

We took care of every minor comment by accordingly modifying the manuscript.

**References**

[1] Gerardo Benito, Marıa José Machado, and Alfredo Pérez-González. "Climate change and flood sensitivity in Spain". In: *Geological Society, London, Special Publications* 115.1 (1996), pp. 85–98.

[2] Cholaw Bueh and Hisashi Nakamura. "Scandinavian pattern and its climatic impact". In: *Quarterly Journal of the Royal Meteorological Society: A journal of the atmospheric sciences, applied meteorology and physical oceanography* 133.629 (2007), pp. 2117–2131.

[3] HJ Fowler and CG Kilsby. "A weather-type approach to analysing water resource drought in the Yorkshire region from 1881 to 1998". In: *Journal of Hydrology* 262.1-4 (2002), pp. 177–192.

[4] Michael Hofstätter et al. "Large-scale heavy precipitation over central Europe and the role of atmospheric cyclone track types". In: *International Journal of Climatology* 38 (2018), e497–e517.

[5] Ian D Phillips and Glenn R McGregor. "The utility of a drought index for assessing the drought hazard in Devon and Cornwall, South West England". In: *Meteorological Applications* 5.4 (1998), pp. 359–372.

[6] Muhammad Rahiz and Mark New. "Spatial coherence of meteorological droughts in the UK since 1914". In: *Area* 44.4 (2012), pp. 400–410.

[7] Faize Sariş, David M Hannah, and Warren J Eastwood. "Spatial variability of precipitation regimes over Turkey". In: *Hydrological Sciences Journal–Journal des Sciences Hydrologiques* 55.2 (2010), pp. 234–249.

[8] Jonathan Spinoni et al. "Meteorological droughts in europe: events and impacts-past trends and future projections". In: (2016).

[9] J-P Vidal et al. "Multilevel and multiscale drought reanalysis over France with the Safran-Isba-Modcou hydrometeorological suite". In: *Hydrology and Earth System Sciences* 14.3 (2010), pp. 459–478.

---

## Author Comment (AC2)

We thank the referee for investing her/his valuable time to help us improve our work with her/his accurate suggestions. We have revised our manuscript, taking into consideration all the referee's comments. The parts in red have been added to the manuscript to improve our work.

**1. In Introduction, the research motivation of the spatial patterns and features of meteorological droughts is not clear enough.**

*We strongly extended the second to last paragraph in the introduction as follows.*

Our main objective here is to uncover spatial features of meteorological droughts in Europe, highlighting underlying mechanisms and patterns which could potentially support drought's forecast in the future. We aim at distinguishing regions in Europe whose main feature lies in drought occurrence and propagation's coherence. Identifying such territories could be of a great importance to further investigate this phenomenon within those areas where its characteristics are homogeneous. Indeed, droughts display a high spatial and temporal variability, and it is thus fundamental to study their evolution accounting for these irregularities to possibly lower uncertainties. Furthermore, with our model we are able to describe the average historical patterns in droughts' evolution which could be a starting point for future climate studies to identify the spatial tracks that are followed by this climate hazard, building a forecasting scheme. Our study is based on climate complex networks and on the concept of Event Synchronization, a nonlinear statistical similarity method useful to determine the correlations among spatial locations in terms of event co-occurrences. Using these tools we are able to identify drought regions in Europe based on the process itself and not depending on any external classifications, bringing out key aspects concerning drought dynamics at a regional scale for different rainfall accumulation periods from 1981 to 2020, while introducing new methodologies in general climate networks reconstruction from raw data. The understanding and ability of describing droughts as a complex phenomenon is still in a preliminary stage, but climate complex networks prove to be a powerful tool to reveal hidden features of this climatic process.

**2. Why is Hamming distance used to calculate the difference between two subnetworks, is there any other distance (measure) can be used.**

Several algorithms have been proposed in graph theory to compare the topology of two networks. Graph similarity metrics are often needed in several applications where different topologies have to be compared (see [6, 12, 13]). Graph edit distances are amongst the most used metrics, aiming at finding the cost of transforming a first graph into another with subsequent edit operations (generally nodes and edges insertions, deletions and substitutions) [7]. One of the issue of this procedure is that the cost of each different edit operation has to be set arbitrarily. A special case in GEDs is the Hamming distance, which measures the structural difference between two graphs only in terms of edges' placement. The reason why we have chosen this metric over the others is threefold: the networks we are comparing are undirected, unweighted and have the same link density. The fact that the networks are undirected and unweighted allows us to compute the Hamming distance by simply counting the times in which an edge is present in one of the graph and does not appear in the other, a procedure which is computationally very easy and which gives an enough informative result. Moreover, the compared graphs have the same link density: we do not need to account for the difference in the sparsity of the networks, which could be done with the Jaccard distance, another structural GED metric [4].

We should also mention that instead of looking at local differences through structural distances, it is possible to use spectral methods to capture the changes of a graph as a whole [8], by investigating the spectral properties of the adjacency matrix. However, we are using the Hamming procedure precisely because we want to minimize the differences in edges' placement and we have no reason to value one specific edge presence more than any other (the Hamming distance treats all edges uniformly); for different purposes, the distance measure has to be chosen to address specific aspects of a network's properties, e.g. information flow, geometrical information, nodes' centrality, etc.

*The following paragraph has been added to the manuscript.*

The Hamming distance is not the only possible choice to compare the topology of two networks [6, 12, 13]. The reason why we have chosen this metric over the others is threefold: the networks we are comparing are undirected, unweighted and have the same link density. For this specific case, this method gives an enough informative result, being also computationally very easy. For more complicated scenarios, one should look into other more refined Graph edit distances [7] or into spectral methods [8] to capture the changes of a graph as a whole.

**3. In the study of the spatial pattern and characteristics of meteorological drought, what are the advantages of using method Synchronization analysis compared to other methods.**

We choose to use Event Synchronization to assess the pairwise similarities between locations in terms of droughts' occurrences. The event-like nature of the phenomenon under study calls for a statistical measure designed to estimate correlations among time series with events defined on them. Moreover, ES is a nonlinear method, able to treat event series with not equal spacing between successive occurrences without fixing a time lag a priori. For these reasons it has become one of the most used method to study extreme events in the climate network's field [1–3, 5, 9–11]. Ultimately, we would summarize the advantages of this method as follows:

- ES is designed to treat event-like time series;

- by using ES there is no need to set a specific time lag;

- ES has both a symmetric and asymmetric formulation, eventually being able to show driver-response relationships (we have used it in both ways in our study);

- ES has been extensively used as a tool to construct climate extreme events' networks, proving to be enough efficient and informative.

*We extended the explanation of ES in the manuscript as follows.*

Event synchronization is a powerful nonlinear method to assess the similarity of event series with not equal spacing between successive occurrences and thus it is especially appropriate for studying extreme events [5]. The degree of synchronicity of two event series is measured based on the relative timings of events and it is obtained from the number of quasi-simultaneous occurrences. We summarize the advantages of this method as follows: (i) ES is designed to treat event-like time series; (ii) by using ES there is no need to set a specific time lag; (iii) ES has both a symmetric and asymmetric formulation, eventually being able to show driver-response relationships; (iv) ES has been extensively used as a tool to construct climate extreme events' networks, proving to be enough efficient and informative [1–3, 5, 9–11]. The detailed algorithm is described in [5] and is shortly repeated in the Appendix for the convenience of the reader.

**References**

[1] Ankit Agarwal et al. "Multi-scale event synchronization analysis for unravelling climate processes: a wavelet-based approach". In: *Nonlinear Processes in Geophysics* 24.4 (2017), pp. 599–611.

[2] Niklas Boers et al. "Complex networks reveal global pattern of extreme-rainfall teleconnections". In: *Nature* 566.7744 (2019), pp. 373–377.

[3] Niklas Boers et al. "Prediction of extreme floods in the eastern Central Andes based on a complex networks approach". In: *Nature communications* 5.1 (2014), p. 5199.

[4] Claire Donnat and Susan Holmes. "Tracking network dynamics: A survey of distances and similarity metrics". In: *arXiv preprint arXiv:1801.07351* (2018).

[5] Jingfang Fan et al. "Statistical physics approaches to the complex Earth system". In: *Physics reports* 896 (2021), pp. 1–84.

[6] Mirtha-Lina Fernández and Gabriel Valiente. "A graph distance metric combining maximum common subgraph and minimum common supergraph". In: *Pattern Recognition Letters* 22.6-7 (2001), pp. 753–758.

[7] Xinbo Gao et al. "A survey of graph edit distance". In: *Pattern Analysis and applications* 13 (2010), pp. 113–129.

[8] Giuseppe Jurman, Roberto Visintainer, and Cesare Furlanello. "An introduction to spectral distances in networks". In: *Proceedings of the 2011 conference on Neural Nets WIRN10: Proceedings of the 20th Italian Workshop on Neural Nets*. 2011, pp. 227–234.

[9] N Malik, N Marwan, and J Kurths. "Spatial structures and directionalities in Monsoonal precipitation over South Asia". In: *Nonlinear Processes in Geophysics* 17.5 (2010), pp. 371–381.

[10] Veronika Stolbova et al. "Topology and seasonal evolution of the network of extreme precipitation over the Indian subcontinent and Sri Lanka". In: *Nonlinear Processes in Geophysics* 21.4 (2014), pp. 901–917.

[11] Felix M Strnad et al. "Extreme rainfall propagation within Boreal Summer Intraseasonal Oscillation modulated by Pacific sea surface temperature". In: *arXiv preprint arXiv:2302.00425* (2023).

[12] Julian R Ullmann. "An algorithm for subgraph isomorphism". In: *Journal of the ACM (JACM)* 23.1 (1976), pp. 31–42.

[13] Laura A Zager and George C Verghese. "Graph similarity scoring and matching". In: *Applied mathematics letters* 21.1 (2008), pp. 86–94.